# 1921–2021: A Century of Renewable Ammonia Synthesis

**Kevin H. R. Rouwenhorst** [1,2,3,*] 🔿, **Anthony S. Travis** [4] **and Leon Lefferts** [1] 🔿

1  Catalytic Processes & Materials, MESA+ Institute for Nanotechnology, Department of Science & Technology, University of Twente, P.O. Box 217, 7500 Enschede, The Netherlands; l.lefferts@utwente.nl
2  Proton Ventures, Karel Doormanweg 5, 3115 Schiedam, The Netherlands
3  Ammonia Energy Association, 77 Sands Street, 6th Floor, Brooklyn, NY 11201, USA
4  Edelstein Center, The Hebrew University of Jerusalem, Jerusalem 91904, Israel; tony.travis282@gmail.com
*  Correspondence: k.h.r.rouwenhorst@utwente.nl

**Abstract:** Synthetic ammonia, manufactured by the Haber–Bosch process and its variants, is the key to securing global food security. Hydrogen is the most important feedstock for all synthetic ammonia processes. Renewable ammonia production relies on hydrogen generated by water electrolysis using electricity generated from hydropower. This was used commercially as early as 1921. In the present work, we discuss how renewable ammonia production subsequently emerged in those countries endowed with abundant hydropower, and in particular in regions with limited or no oil, gas, and coal deposits. Thus, renewable ammonia played an important role in national food security for countries without fossil fuel resources until after the mid-20th century. For economic reasons, renewable ammonia production declined from the 1960s onward in favor of fossil-based ammonia production. However, renewable ammonia has recently gained traction again as an energy vector. It is an important component of the rapidly emerging hydrogen economy. Renewable ammonia will probably play a significant role in maintaining national and global energy and food security during the 21st century.

**Keywords:** ammonia; renewable; electrolysis; Haber–Bosch; hydropower; fertilizers; energy

## 1. Introduction

The synthesis of ammonia ($NH_3$) from unreactive nitrogen ($N_2$) and hydrogen ($H_2$), Equation (1), is one of the most significant scientific–technical developments in human history. Ammonia, as the precursor for various nitrogen-containing fertilizers, currently sustains about half of the global population [1,2]. With new strains of wheat and rice, its bulk availability has, particularly since the 1960s, enabled an alleviation of widespread famine [3]. The ammonia synthesis method was invented by Fritz Haber and Robert Le Rossignol in 1909, and was scaled up by Carl Bosch and colleagues at the German firm BASF, which opened the first commercial plant in 1913 at Oppau, near its Ludwigshafen works [4]. The hydrogen was produced from a coal-based process.

$$3H_2 + N_2 \rightleftharpoons 2NH_3 \text{ with } \Delta Hr = -46 \text{ kJ mol-NH}_3^{-1}, \tag{1}$$

However, 100 years ago, in 1921, the first viable commercial rival to what became known as the Haber–Bosch process was introduced by the Italian chemist Luigi Casale. His synthetic ammonia process originally relied on hydrogen generated by the electrolysis of water. The Casale ammonia synthesis technology became the key to the globalization of the synthetic ammonia industry.

Currently, about 183 Mt of ammonia is produced annually, almost exclusively from fossil feedstocks, mainly natural gas and coal [5,6]. Renewable ammonia production today accounts for just 0.01% of global production [7]. It is estimated that current ammonia production accounts for 0.5 Gt of $CO_2$-equivalent emissions annually [8], equivalent to 1.0% of global $CO_2$-equivalent emissions.

Renewable ammonia production has recently gained traction again, due to its potential role as a decarbonized hydrogen carrier and as a fuel in the hydrogen economy. Historically, nearly all electrolysis-based hydrogen production capacity has been used for ammonia synthesis [9], making any discussion regarding renewable ammonia in the context of the hydrogen economy relevant. Hydropower has historically been the renewable electricity source for alkaline electrolyzers for hydrogen production [10].

In this paper, we discuss the historical commercialization of renewable ammonia synthesis from 1921, its scale up in the late 1920s and beyond, and the demise of renewable ammonia during the second half of the 20th century (1960s–2021). To our knowledge, this is the first publication specifically discussing the history of renewable ammonia. We use recent literature, as well as sources from the early 20th century, covering a century of renewable ammonia production.

The historical role of renewable ammonia synthesis is demonstrated in Figure 1. The only renewable ammonia plant still in operation is located in Cuzco, Peru. However, various renewable ammonia plants have recently been announced, mainly based on solar and wind combined with electrolysis [7].

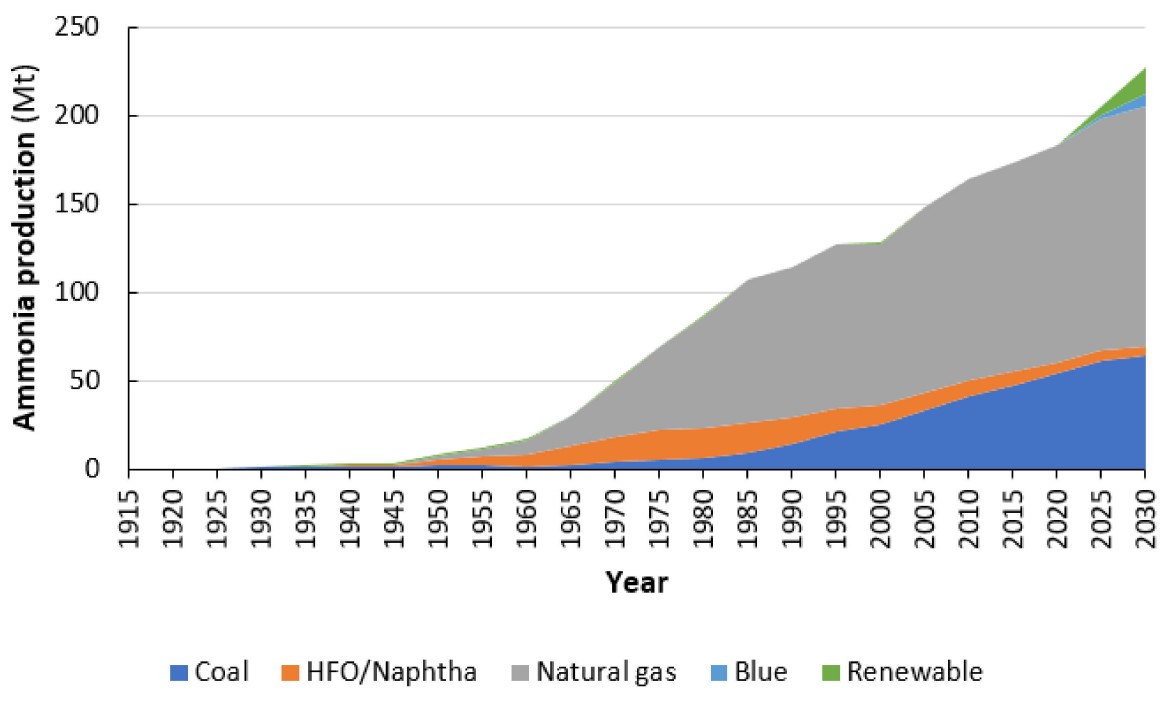

**Figure 1.** *Cont.*

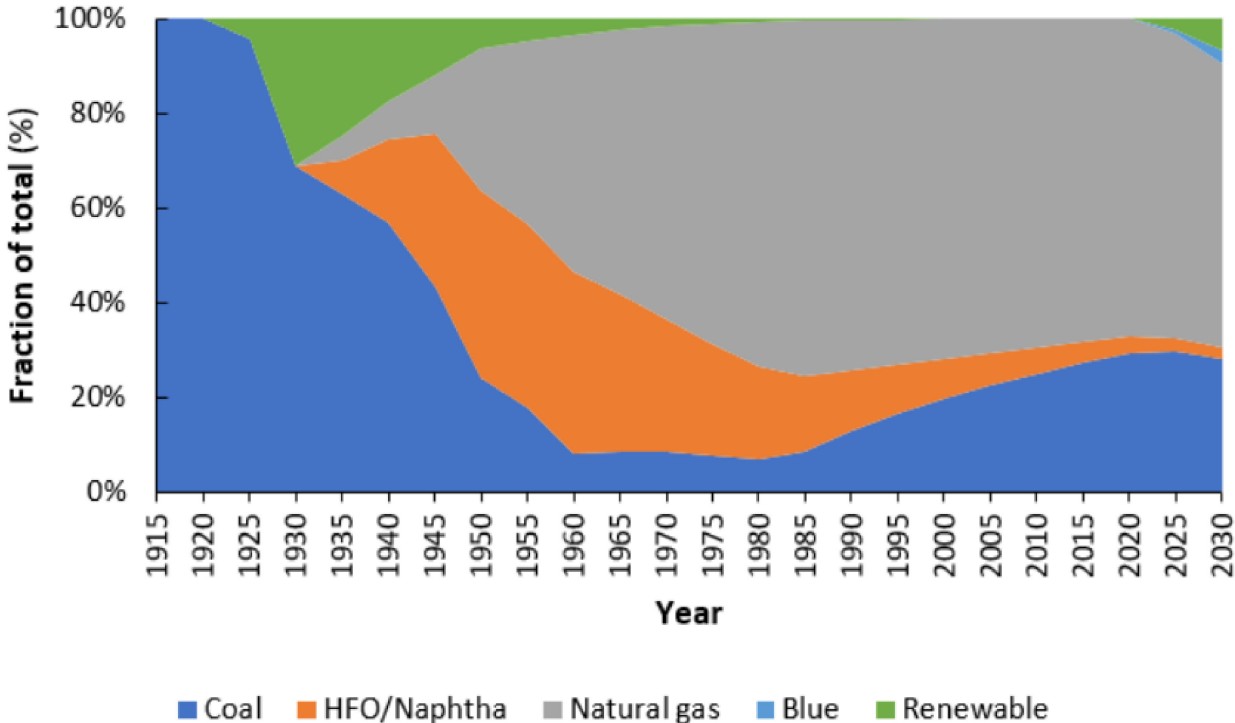

**Figure 1.** Historical ammonia production by feedstock, and expected production until 2030. Coal: coal gasification; HFO: heavy fuel oil gasification; Naphtha: naphtha reforming; Natural gas: steam methane reforming; Blue: steam methane reforming with carbon capture and storage (CCS); Renewable: electrolysis. The original data can be found in the Supplementary Materials Section.

## 2. Early 1920s: Development and Small-Scale Technology

In 1913, the first ammonia plant began operation at the Oppau works of BASF, in Germany. From 1913 until 1920, ammonia was synthesized only in Germany, based on the BASF Haber–Bosch process, which used coal-based technology for gas production. Subsequently, ammonia synthesis technology was developed outside Germany. From 1921, electrolysis-based hydrogen production was a relevant technology for ammonia synthesis [10–12]. However, as at BASF, the majority of ammonia produced by similar processes in the 1920s was synthesized from hydrogen produced by coal gasification and coking oven processes [10]. By 1930, about 30% of the total ammonia production capacity was based on electrolysis-based ammonia synthesis with individual ammonia unit capacities up to 295 t-NH$_3$ d$^{-1}$ (at Rjukan, Norway, see Supplementary Materials Section). This was the maximum rated capacity of individual ammonia units (synthesis loops) until the mid-1960s, when centrifugal compressors were introduced. Hydrogen for ammonia was also produced as a by-product from other electrochemical processes, such as caustic or chlorine production [10], although production volumes for these plants were limited to ≤10 t-NH$_3$ d$^{-1}$.

Electrolysis-based ammonia production was mainly developed in locations with cheap and abundant electricity from hydropower [12]. In fact, most of the regions that adopted electrolysis-based ammonia synthesis already had substantial hydropower capacity that was used for the fixation of atmospheric nitrogen for use in fertilizer production. Before the development of the ammonia synthesis, nitrogen was industrially fixed with the Birkeland–Eyde electric arc process and the Frank–Caro calcium cyanamide process, both dating from 1905, and both of which consumed hydropower. The Birkeland–Eyde process fixed nitrogen from the air by reacting atmospheric nitrogen (N$_2$) and oxygen (O$_2$) in a plasma reactor, thereby forming nitrogen oxides (NO$_X$), which were treated with water to form nitric acid (HNO$_3$) [13,14]. The Frank–Caro process fixed nitrogen from air by reacting

atmospheric nitrogen (N$_2$) with calcium carbide (CaC$_2$), thereby forming the final product calcium cyanamide (CaCN$_2$), and the by-product carbon (C) [10].

Because electrolysis-based ammonia synthesis was substantially more energy efficient than the Birkeland–Eyde process and had a lower capital investment than the Birkeland–Eyde and Frank–Caro processes, the ammonia synthesis technology eventually replaced the Birkeland–Eyde and Frank–Caro processes in locations with hydropower capacity [10,12]. The Birkeland–Eyde process, the Frank–Caro process, and the electrolysis-based synthetic ammonia process consumed about 3.1, 0.7, and 0.8 MJ mol-N$^{-1}$, respectively, during the 1920s [10]. The low-temperature electrolysis-based hydrogen production used in the ammonia process has in recent decades been further optimized to yield a current energy consumption of about 0.6 MJ mol-N$^{-1}$ [7].

### 2.1. Renewable Ammonia Synthesis Technology

A schematic overview of electrolysis-based ammonia synthesis during the 1920s is shown in Figure 2. At that time, electrolysis-based ammonia synthesis was the technology with the lowest energy consumption for ammonia synthesis, at about 48–50 GJ t-NH$_3$$^{-1}$ [10,11], and far lower than coal-based ammonia synthesis at about 95–100 GJ t-NH$_3$$^{-1}$ [5]. However, coal and coke were readily available at a low cost in various countries, for example in Germany, Belgium, France, the Netherlands, and the United Kingdom [12,15–18], making fossil-based ammonia production in these countries more economical than renewable ammonia production. Fossil-based processes for hydrogen production required extensive purification equipment, making necessary substantial capital investment.

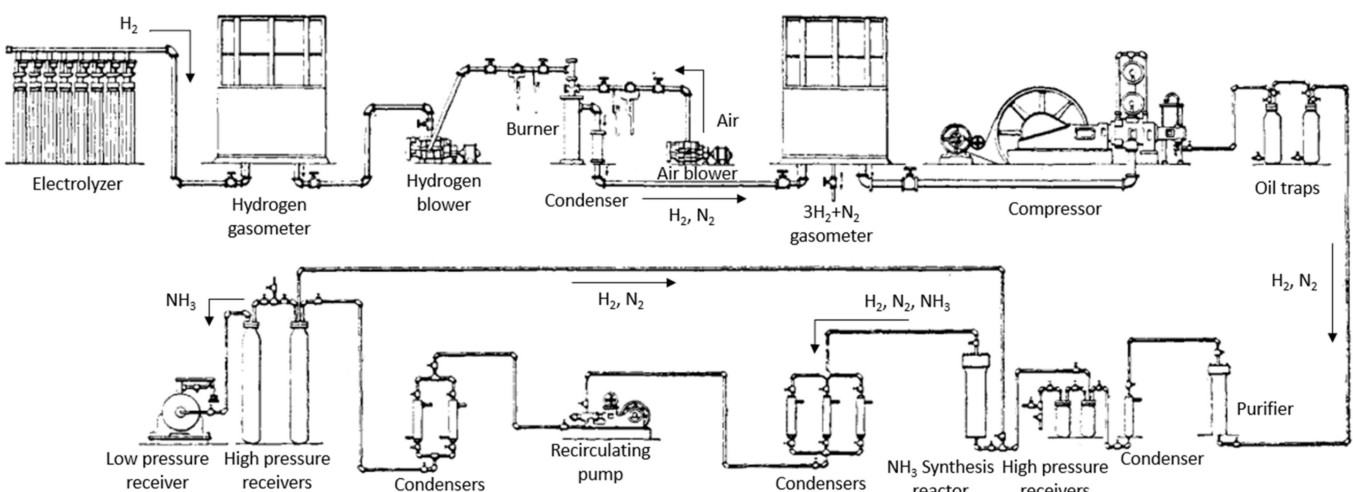

**Figure 2.** Schematic overview of an electrolysis-based ammonia synthesis plant. Reprinted with permission from Ref. [19]. 1925, ACS Publications.

Renewable hydrogen was mainly produced by water splitting, that is, electrolysis. Hydroelectric power plants were coupled with alkaline electrolyzers for producing pure hydrogen. The 1920s electrolyzers typically consumed about 5.2 kWh Nm$^{-3}$ H$_2$ [11]. Since then, electrolyzers have been optimized to an energy consumption of about 4.2–4.7 kWh Nm$^{-3}$ H$_2$ [20,21]. As mentioned earlier, hydrogen used in the synthesis of ammonia was also produced as a by-product from other electrochemical processes, such as caustic alkali or chlorine production [22].

In the renewable processes, nitrogen was obtained from air by cryogenic distillation, or by removing oxygen by burning part of the hydrogen with air to produce water, which was subsequently removed by condensation [19]. The latter technology was only economic at small-scale operations (<5 t-NH$_3$ d$^{-1}$). Nitrogen production by cryogenic air separation is the preferred technology for large-scale electrolysis-based ammonia synthesis [23].

Ammonia is synthesized at considerably elevated temperatures and under high pressures in the presence of an iron-based catalyst which contained promoters. Currently, this technology is commonly termed the Haber–Bosch process. However, various ammonia processes were developed and introduced during the 1920s, following the refusal of the German company BASF to share the Haber–Bosch technology knowhow with foreign firms and governments [12,16]. The alternative ammonia synthesis technologies include the already mentioned Casale process, the Claude process, the Fauser process, the General Chemical/Allied process, the NEC (Nitrogen Engineering Corporation) process, the Mont Cenis process, and the Showa Fertilizer process. Various sources discuss the subtle differences regarding operating conditions and yields among the processes [11,12,18,24]. The different ammonia synthesis technologies are listed in Table 1.

It should be noted that hydrogen production typically accounts for more than 90% of the required energy input. Furthermore, the compressors for feed gas compression and the recycling of unreacted gas in the ammonia synthesis loop can be operated with renewable electricity. If renewable electricity is the economical method for hydrogen production, it will also be favorable for gas compression and recirculation in the synthesis loop. Thus, ammonia production using renewable hydrogen results in renewable ammonia. It should be noted that the implicit assumption that the produced ammonia converts completely back to $N_2$ is reasonable when using ammonia as an energy carrier.

From a thermodynamic point of view, ammonia synthesis benefits from a low temperature and a high pressure. However, the $H_2$ and $N_2$ do not spontaneously react to form ammonia unless the temperature is increased to several thousand Kelvin. This is impracticable in industry. Therefore, a catalyst is essential to increase the ammonia synthesis rate for industrial application.

All synthetic ammonia technologies developed during the 1920s relied on a multi-component iron-based catalyst, high temperatures and high pressures (400–650 °C and 200–1000 bar), and ammonia removal by condensation. The exact formulation of the iron-based catalyst varied among ammonia synthesis processes because most companies developed their own catalysts due to a lack of international collaboration [12]. For this reason, there was a certain amount of industrial espionage to obtain intellectual property. For example, this enabled British investigators to gain access to details of the BASF technology, including catalyst recipes.

The catalyst formulation has a major impact on the activity. A less active catalyst requires a higher operating temperature to achieve sufficient activity for ammonia formation. However, a higher temperature is not beneficial for the equilibrium, as explained above. Thus, the pressure is increased to improve the equilibrium conversion to ammonia.

Most processes, including the original Haber–Bosch process, required refrigeration to sub-atmospheric temperatures to produce liquid, anhydrous ammonia. However, the Casale and Claude processes did not require such refrigeration to sub-atmospheric temperatures to produce liquid, anhydrous ammonia due to their very high operating pressures (800–1000 bar). Thus, the operating pressure influences both the thermodynamic equilibrium and the liquefaction temperature. It should be noted that increasing the pressure increases the energy requirement for the compression of the hydrogen and nitrogen feedstock.

Apart from the (original) Claude process, which was not continuous and did not incorporate a gas recycle, the maximum yield of ammonia at a single pass was no more than around 20% (see Table 1). The unconverted nitrogen and hydrogen were recycled to the ammonia synthesis reactor with the continuous addition of new feed hydrogen & nitrogen.

An exception, as mentioned, was the Claude process, which operated at extreme pressures of 900–1000 bar to bring about the ammonia synthesis in a series of reactors without the recirculation of unreacted nitrogen and hydrogen. Some 40% conversion was achieved in the first pass [11,24–26]. After the removal of the ammonia by condensation, the unconverted feedstock was fed successively to two more reactors, bringing the total yield for three reactors to around 85%. At first, issues with the steel reactors due to the extreme pressures delayed the introduction of the Claude process until the late 1920s [12,16].

From around 1940, the Claude process was redesigned as a continuous process, with a recirculation of unreacted gas, similar to the other ammonia processes.

**Table 1.** Comparison of reported yields for synthetic ammonia technologies. Values adapted from references [12,27].

| Process | Year | Temperature (°C) | Pressure (atm) | Single Pass Conversion (%) |
|---|---|---|---|---|
| Haber–Bosch (Germany) * | 1913 | 550 | 200 | 7–8 |
| Casale (Italy) | 1921 | 500 | 800–850 | 15–18 |
| Claude (France) | 1921–1922 | 500–650 | 900–1000 | 40 ** |
| Fauser (Italy) | 1921–1922 | 500 | 250–300 | 12–23 |
| General Chemical/Allied (United States) | 1921 | 500 | 200 | 20–22 *** |
| Nitrogen Engineering Corporation (United States) | 1926 | 500 | 200–300 | 20–22 *** |
| Mont Cenis (France) | 1925–1926 | 400–425 | 100 | 9–20 *** |
| Showa Fertilizer (Japan) | 1931 | - | - | - |

* Later IG Farben. ** for the first of a series of converters; overall conversion after 3–4 converters about 85–90%. *** Claimed in 1940s, though probably on the high side [28].

In the following sections, the development of renewable ammonia production in various countries during the early 1920s is discussed. Historical accounts of the ammonia industry before the 1940s have been reported by Ernst [10,22], Travis [12,16], and Van Rooij [18]. These authors discussed the ammonia industry in general, mainly focusing on fossil-based ammonia, in contrast to the current paper which focuses on renewable ammonia production.

### 2.2. Italy

At the turn of the 20th century, the Italian chemical industry was poorly developed, due to limited coal resources [16]. Furthermore, there was limited arable land in Italy [16]. Thus, fertilizers were imported to increase crop yields. These included Chile saltpetre from Latin America and ammonium sulfate from Great Britain [16].

Hydroelectric power was developed in Italy by 1900 [16], enabling the development of a chemical industry based on electrochemical processes. In 1905, the first Frank–Caro calcium cyanamide plant was opened in Italy [16]. Various hydropower-driven cyanamide plants were built in Italy prior to and during the 1920s.

Though cyanamide production remained significant in Italy, the main growth was in renewable ammonia plants. In fact, the very first large-scale electrolysis-based ammonia plants were located in Italy. Ammonia was mainly produced with Fauser ammonia synthesis technology (see Table 1) by the Montecatini corporation, which monopolized the nitrogen fertilizer industry in Italy. However, there were smaller contributions from the Casale and Claude processes (see Table 2). Significantly, the ammonia synthesis technologies of Casale and Fauser were both developed in Italy [16]. Moreover, the technology for electrolysis-based hydrogen production was invented by both Casale and Fauser. The Fauser process was restricted to Italy until 1926, while the Casale process was licensed worldwide from 1921.

**Table 2.** Renewable ammonia plants in Italy. Values adapted from references [12,22,29].

| Location | Ammonia Technology | Year | Capacity (kt-NH$_3$ y$^{-1}$) |
|---|---|---|---|
| Terni/Nera Montoro * | Casale | 1921–1923 | 0.7 |
| | | 1924 | 2.6 |
| | | 1926 | 7.0 |
| | | 1927 | 10.5 |
| Bussi | Claude | 1923 ** | 2.5 |
| Belluno (Mas) | Fauser | 1924 | - |
| Merano | Fauser | 1925 | 37.8 |
| | | 1930s | 35.0 |
| Agordo | - | 1926 *** | 3.5 |
| Novara | Fauser | 1926 | 7.5 |
| Coghinas | Fauser | 1927 | 3.5–7.0 |
| Crotone | Fauser | 1927 | 7.0 |
| | | 1930s | 24.5 |
| San Giuseppe di Cairo | Fauser | N/A | 17.5 |
| | | 1930s | 44.8 |
| Taranto | Fauser | - | - |
| Massina | Fauser | - | - |

* Nera Montoro is close to Terni, see Figure 2. ** Year unknown, but a partnership for the construction was agreed by 1923. *** Year unknown, but in operation by 1926.

The first renewable ammonia plant was built in Terni, Italy. This semi-commercial plant was based on the electrolysis technology and the ammonia synthesis technology of Luigi Casale [12,29]. By the autumn of 1922, the plant produced about 2 t-NH$_3$ d$^{-1}$, equivalent to 0.7 kt-NH$_3$ y$^{-1}$ [12]. The Terni plant, mainly used for R&D purposes, was expanded in 1923 and 1924, to produce about 7–8 t-NH$_3$ d$^{-1}$ [12]. Due to a higher ammonia production in other locations by the late 1920s (see Table 2), especially as a result of Montecatini's aforementioned monopoly over nitrogen fertilizer production in Italy, Terni declined as an ammonia production facility and from 1925 became a leading research center for Casale's company [16]. Casale ammonia technology was used close to the hydro power station in Nera Montoro (see Figure 3).

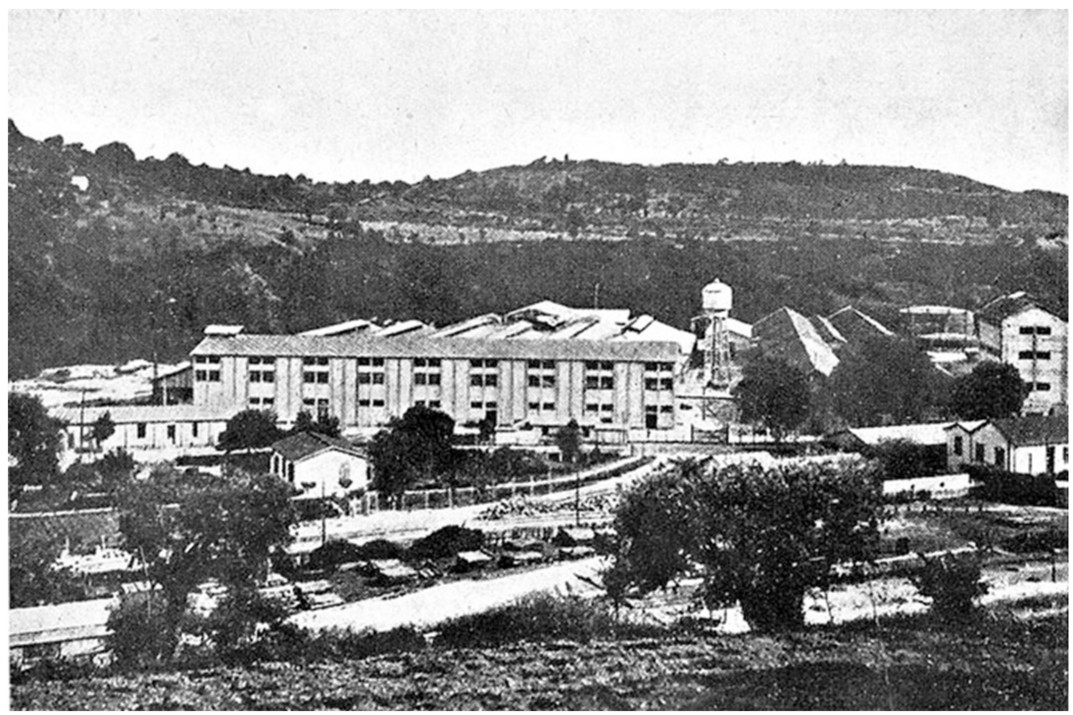

**Figure 3.** Ammonia factory at Nera Montoro, Italy. Courtesy Casale S.A.

Most important to this discussion is the Fauser–Montecatini renewable ammonia plant in Merano, South Tyrol, located next to the Marlengo hydroelectric power station. This plant produced about 108 t-NH$_3$ d$^{-1}$ by 1925. It was the largest ammonia plant outside Germany and at the time the largest renewable ammonia plant in the world. Under the state-sponsored program of Mussolini, the chemical industry was a key pillar for the fascist government in Italy at the time, and ammonia synthesis grew rapidly between 1925 and 1927 [16].

Renewable ammonia production was not restricted to mainland Italy. In 1927, an ammonia plant was built on the island of Sardinia, next to a hydroelectric power plant across the River Coghinas. The plant was, again, based on Fauser technology and had a production capacity of 10 t-NH$_3$ d$^{-1}$.

Numerous other renewable ammonia plants were constructed in Italy, mainly based on Fauser technology. The Italian ammonia plants are listed in Table 2. This clearly demonstrates that Italy has had the most renewable ammonia plants built to date.

### 2.3. Other European Countries

During the early 1920s, various renewable ammonia plants were constructed in other European countries, such as France, Spain, Sweden, and Switzerland. The renewable synthetic ammonia plant at Rjukan, Norway, was constructed later, and is discussed in Section 3.1.

In Saint Auban, France, a renewable ammonia plant with Casale ammonia synthesis technology was constructed sometime after 1922. In 1927, this plant was expanded to a daily capacity of about 10 t-NH$_3$ d$^{-1}$ [22].

In 1925, a 5 kt-NH$_3$ y$^{-1}$ plant was constructed in Sabiñánigo, Spain, again based on Casale ammonia synthesis technology (see Figure 4) [12]. Another ammonia plant with Casale technology was opened in Visp, Switzerland, in 1927. Casale's 3.5 t-NH$_3$ d$^{-1}$ converter gave an actual production of about 4 t-NH$_3$ d$^{-1}$ [12]. Thus, the annual capacity was about 2.8 kt-NH$_3$ y$^{-1}$, based on two converters. Both plants relied on electrolysis for hydrogen. Casale converters of different-rated capacities of up to 20 t-NH$_3$ d$^{-1}$ were available by the late-1920s.

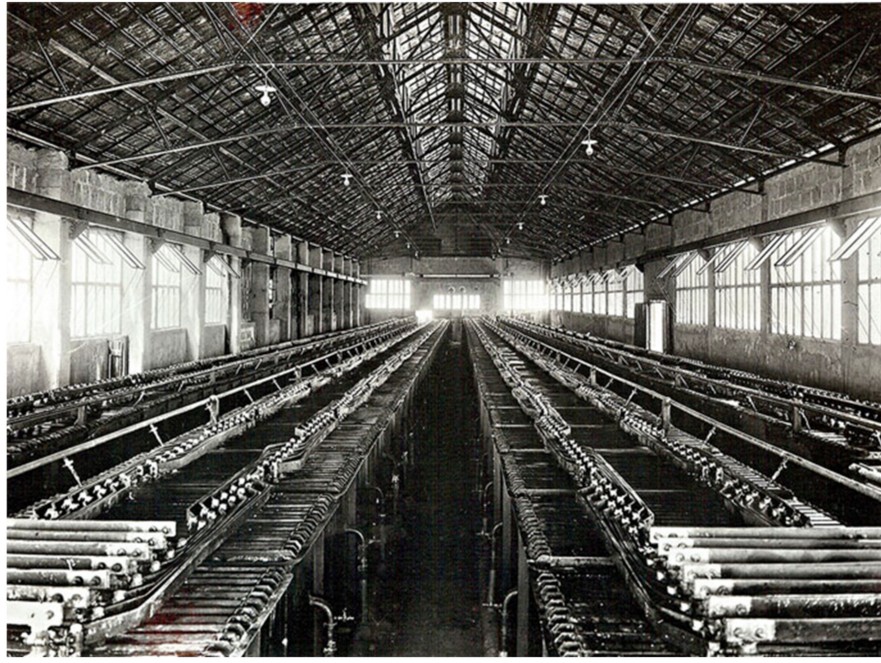

**Figure 4.** Electrolyzers in the Sabiñánigo ammonia factory in Spain, constructed in 1925. Courtesy Casale S.A.

In Ljunga, Sweden, the Stockholms Superfosfat Fabrik replaced its cyanamide process with a renewable ammonia plant based on Fauser ammonia synthesis technology [10]. Finally, a government-owned renewable ammonia plant based on Casale technology with a capacity of 11 t-NH$_3$ d$^{-1}$ was located at Dugi Rat in Yugoslavia [12].

*2.4. Japan*

Like Italy, Japan had scarce arable land but abundant water power. It also developed hydroelectric power around the turn of the 20$^{th}$ century [16]. During the early 1900s, the main nitrogen fertilizers were imported bean cake and ammonium sulfate. However, this nitrogen fertilizer supply was gradually supplemented by calcium cyanamide plants in Japan that were powered with hydropower, starting in Minamata in 1909 [16]. The calcium cyanamide had a dirty, dusty appearance, and farmers believed this would poison crops. Therefore, the calcium cyanamide was converted to ammonia, which was reacted with sulfuric acid to produce the fertilizer ammonium sulfate [10,16].

Three Japanese delegations had travelled to Italy in 1921 to see the electrolysis-based ammonia plant based on Casale technology in operation (see Section 2.2). As a result, Japan became the first non-European country to build an electrolysis-based synthetic ammonia plant, based on Casale technology.

In late 1923, 7 kt-NH$_3$ y$^{-1}$ Casale ammonia units were introduced at Nobeoka (see Figure 5) [12]. At the time this was the largest renewable ammonia plant worldwide. The plant was later expanded in 1927 with new Casale converters. By then, the plant produced about 19.5 kt-NH$_3$ y$^{-1}$ [22]. Other estimates include a total rated plant capacity, based on larger converters, of 62 t-NH$_3$ d$^{-1}$ [16].

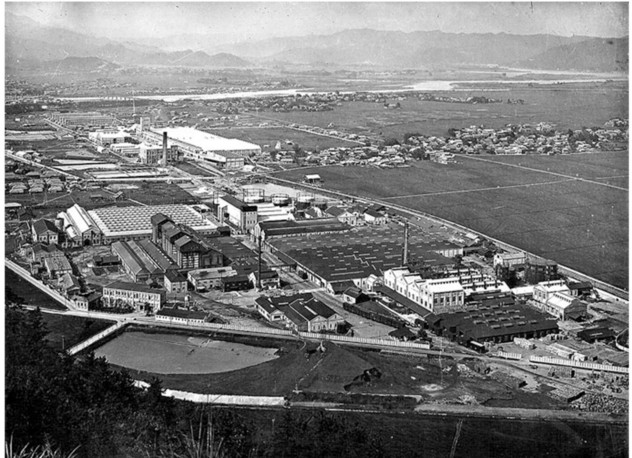 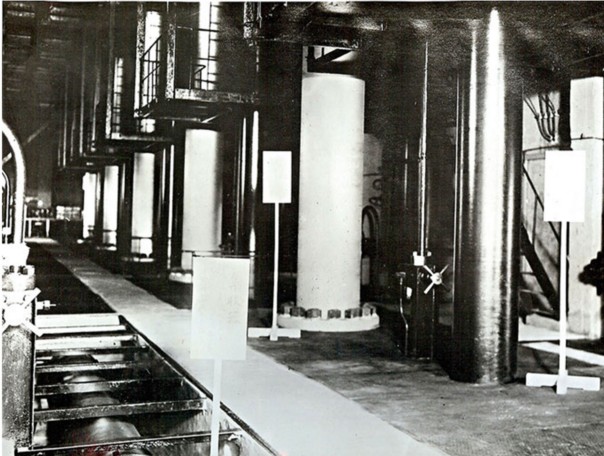

**Figure 5.** Left: Nobeoka ammonia factory in Japan, around 1924. Courtesy Casale S.A. Right: Ammonia converters at Minamata, Japan. Courtesy Casale S.A.

In 1926, an even larger Japanese ammonia plant was built with a capacity of 21.5–32 kt-NH$_3$ y$^{-1}$ in Minamata, also based on Casale ammonia synthesis technology [12,22]. Capacity was later increased to 100 t-NH$_3$ d$^{-1}$. Casale ammonia converters at Minimata are shown in Figure 5.

State sponsored programs for the chemical industry were key aspects of colonial modernization, and Japan was no exception [16]. The Japanese Empire annexed Korea by 1910, and between 1927 and the early 1930s, two large Casale-based renewable synthetic ammonia plants were also constructed there. In Hungnam in the northeast (currently North Korea), an ammonia plant with a capacity of 120 t-NH$_3$ d$^{-1}$ began operation in 1930, utilizing hydroelectric power [16]. By 1938, the ammonia capacity of this plant and the second nearby plant, both owned by Nitchitsu of Japan (operator of Nobeoka and Minamata), was 480 t-NH$_3$ d$^{-1}$.

In 1934, Japan was the third largest producer of fixed nitrogen after Germany and the United States, with an annual output of 200 kt ammonia [16]. The Haber–Bosch technology of IG Farben was introduced in Japan from the mid-1930s onwards, after the initial successes of the Casale processes and the failings of the Claude process. Showa Fertilizers operated a Japanese-designed synthetic ammonia process. During the Second World War, various Japanese ammonia factories were destroyed by the US Army Air Force; from 1945, the Red Army removed equipment from the factory in Hungnam [16]. The Hungnam factory was a target during the Korean War, but was later rebuilt with Chinese and Eastern Bloc assistance.

*2.5. Canada*

Canada had operational renewable nitrogen-fixation technology as early as 1903, with the short-lived Bradley and Lovejoy process, a plasma process similar to the Birkeland–Eyde process, for $NO_X$ production located in Niagara Falls, with a power supply of about 2.2 kW [10]. In practice, this plant was a failure. This plasma-based nitrogen-fixation technology had a high energy consumption, and the emergence of the Frank–Caro process and later the Haber–Bosch process brought about the demise of the Birkeland–Eyde process [12,13]. By the late 1920s, the American Cyanamid Company produced about 54 kt-N $y^{-1}$ of calcium cyanamide with the Frank–Caro process on the Canadian side of the Niagara Falls [30]. This facility opened around 1910.

Nitrogen-fixation capacity for synthetic ammonia was installed in two sites in Canada in 1930. A renewable ammonia plant with Casale technology started operating in Sandwich, Ontario, with an output of about 2.5 kt-$NH_3$ $y^{-1}$. A 38 kt-$NH_3$ $y^{-1}$ facility based on Fauser technology, with hydrogen also obtained by electrolysis, was built in Trail, British Columbia, at the forerunner of the company Cominco [12]. Later, this plant had a reported capacity of 200 t-$NH_3$ $d^{-1}$, equivalent to about 70 kt-$NH_3$ $y^{-1}$ [31]. It was located near a hydroelectric power station on the Kootenay River. Initially, this plant used unipolar electrolyzers from three different companies, namely Fauser, Pechkranz, and Knowles & Stewart [31]. Later, all these electrolyzers were replaced by those supplied by Cominco [31]. The main growth in the Canadian synthetic ammonia industry took place from around 1950, and was based on natural gas. Electrolysis-based hydrogen production at the Trail plant was abandoned after the oil crisis of the 1970s, when the export of hydroelectric energy to the USA became feasible [31]. The feedstock was switched to natural gas, which was abundant on the west coast of Canada [31].

*2.6. United States*

The United States is unique regarding the renewable hydrogen feedstock for ammonia production. Renewable ammonia at first was limited to mainly small units of a rated capacity of 3 t-$NH_3$ $d^{-1}$ designed by the government's Fixed Nitrogen Research Laboratory (FNRL), and drawing on the by-product hydrogen. Two renewable ammonia production facilities were installed on the US side of the Niagara Falls. There, Roessler & Hasslacher Chemical Company operated a FNRL 3 t-$NH_3$ $d^{-1}$ unit with a hydrogen by-product from electrolysis [10]. Later, Roessler & Hasslacher erected a 6 t-$NH_3$ $d^{-1}$ plant, using two FNRL units, to utilize by-product hydrogen available from the Hooker Electrochemical Company. Similarly, Mathieson Alkali Company operated a 10–12 t-$NH_3$ $d^{-1}$ plant, again based on FNRL units, at Niagara Falls, from by-product hydrogen. In Pittsburg, California, by-product hydrogen from chlorine manufacture was used by Great Western Electrochemical to produce about 1 t-$NH_3$ $d^{-1}$ [22]. During 1924–1927, the Swiss Ammonia Casale Company (now Casale S.A.) operated a renewable ammonia facility at Niagara Falls. Its Electrolab corporation's electrolyzers for renewable hydrogen production were transferred to the renewable ammonia plant of Pacific Nitrogen in Seattle. By-product hydrogen was used by General Chemical/Allied at Solvay's Syracuse works for a semi-commercial operation, from 1927 until the early 1930s.

The renewable ammonia produced in the United States, however, was also derived from biomass gasification with the subsequent formation of ammonia from hydrogen and nitrogen [22]. A biomass-based ammonia plant was located in Peoria, Illinois (Commercial Solvents), producing about 12.3 t-NH$_3$ d$^{-1}$ by the end of 1926 [22]. In 1927, a similar plant was constructed in Terre Haute, Indiana, with a capacity of 12–15 t-NH$_3$ d$^{-1}$ [10]. The hydrogen was produced as a by-product from corn fermentation. However, these plants, though successful, produced ammonia for less than a year. The converters were used to produce methanol instead from 1928 onwards [12].

During the 2000s and early 2010s, interest in biomass-based ammonia production re-emerged as an alternative for renewable ammonia production in the United States [32]. However, these plants have not materialized so far.

### 3. Late 1920s–1960s: Scale-Up of Renewable Ammonia and Competition from Fossil Technology

As mentioned in the foregoing, electrolysis-based hydrogen production for ammonia synthesis generally replaced nitrogen fixation by the Birkeland–Eyde process and the Frank–Caro process. This was due to the lower energy consumption of ammonia synthesis compared to the other processes. During the early 1920s, the feasibility of operating electrolysis-based ammonia was established. As of 1930, electrolysis-based hydrogen accounted for about 30% of global ammonia production.

The general interest in synthetic ammonia stimulated the scale-up of ammonia synthesis from a few tons of ammonia per day to hundreds of tons of ammonia per day in the decades that followed. By the late 1920s, a small-scale plant operating at the rate of a few tons of ammonia per day, based on electrolysis, or other feedstocks, was no longer competitive with the larger fossil-based plants [10]. This was especially true during periods of fluctuation in demand, including overcapacity of ammonia, at first at the end of the 1920s [16]. Nevertheless, until well after 1945, a number of large, more cost-competitive renewable ammonia plants were built in various countries, including Egypt, Iceland, India, Norway, Peru, and Zimbabwe [9], in order to benefit from economies of scale. The operational global renewable ammonia capacity over time is shown in Figure 6.

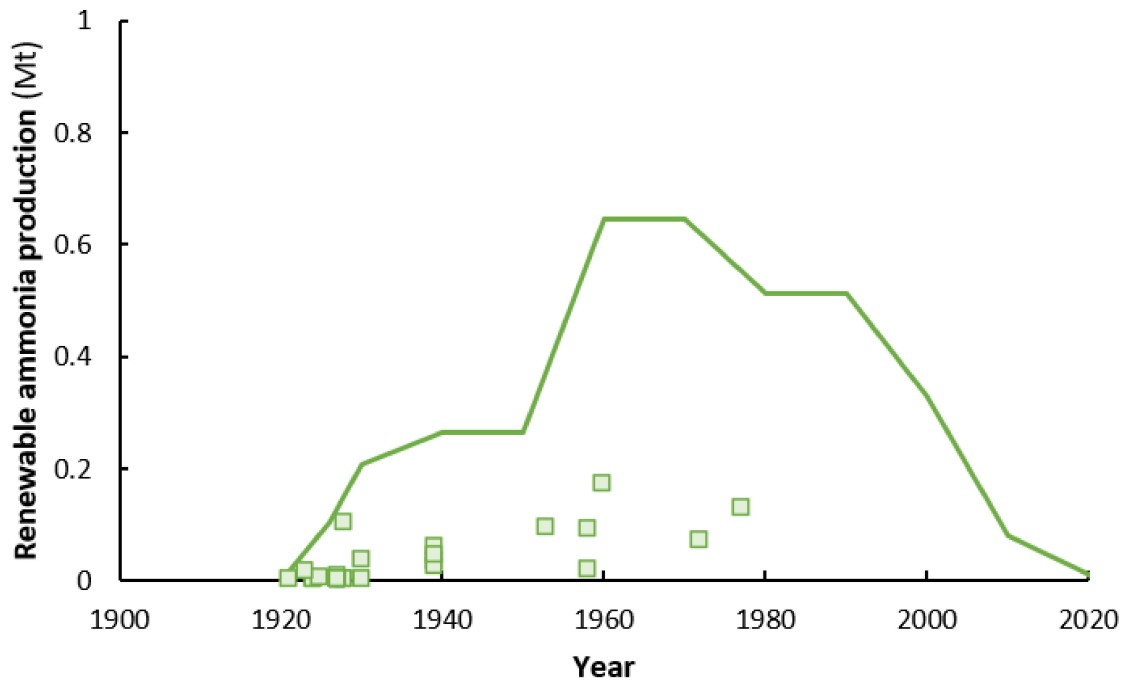

**Figure 6.** Global operational renewable ammonia capacity. Squares represent individual plants (see Table 2 and Table S2).

As renewable ammonia capacity increased, so too did the requirement for electrolyzers. Furthermore, electrolyzers needed to be replaced within 5–10 years [21], while the renewable ammonia plants typically operated for several decades. As a result, various companies started developing and producing electrolyzers and electrolyzer performances improved over time [33,34]. The energy consumption for renewable ammonia synthesis decreased from 48–50 GJ t-NH$_3$$^{-1}$ during the 1920s [10,11] to 36 GJ t-NH$_3$$^{-1}$ during the 1980s [35].

### 3.1. Norway

Norway was one of the first countries to introduce industrial nitrogen fixation. In fact, the Birkeland–Eyde process was developed in Norway by the scientist Kristian Birkeland and the industrialist Samuel Eyde [36,37]. In Notodden, hydroelectric power was fed to electric arc furnaces to fix nitrogen from air with plasma [37]. In 1905, the plant had three 500 kW electric arc furnaces, resulting in an annual capacity of 2 kt calcium nitrate [12]. Various other factories were established by Norsk Hydro in the following years. By 1927, Norway was the country with the largest Birkeland–Eyde process capacity, capable of fixating about 42 kt-N y$^{-1}$ [30].

Instead of transitioning to the Frank–Caro process, Norway changed to electrolysis-based ammonia synthesis in the late 1920s [10]. At the Rjukan site, which already had Birkeland–Eyde furnaces, a pilot plant for ammonia synthesis was operational by 1927, and a year later the plant was working at full capacity. The ammonia synthesis technology was provided by the BASF successor IG Farben, that is the original Haber–Bosch ammonia synthesis technology [12]. With a capacity of about 295–340 t-NH$_3$ d$^{-1}$, the Rjukan plant was nearly triple the size of the Merano plant in Italy (see Section 2.2), which was previously the largest renewable ammonia plant. The electrolyzers were provided by Norsk Hydro (see Figure 7). The plant remained in operation until 1971.

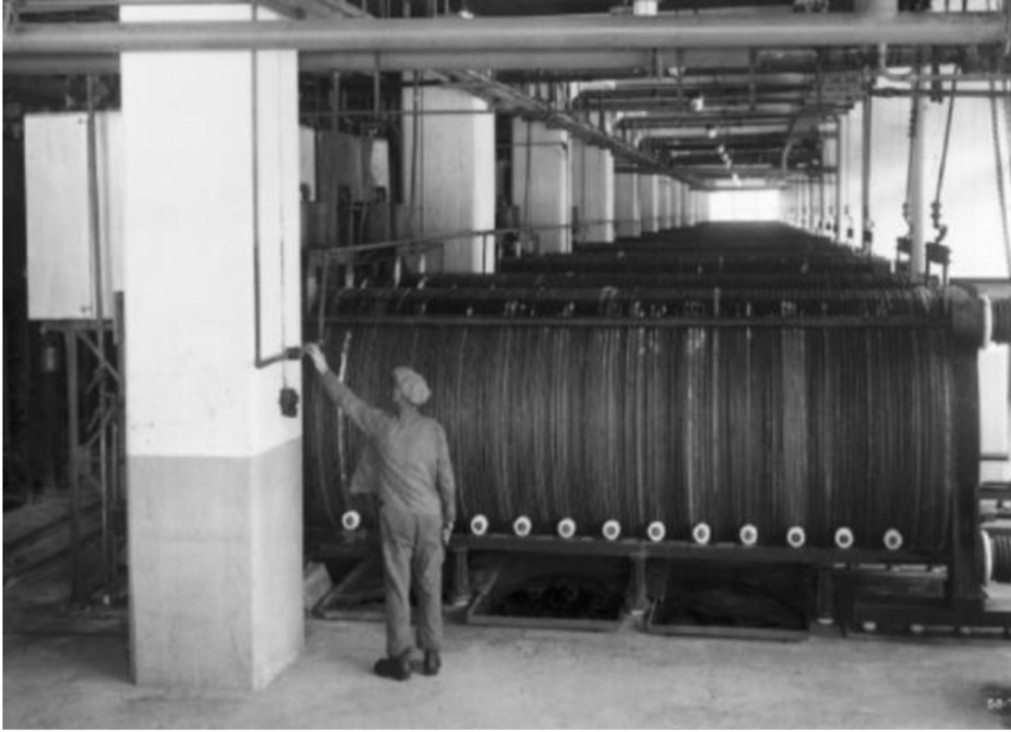

**Figure 7.** Electrolyzers at the Rjukan ammonia factory in Norway, operational 1928–1971. Courtesy Norsk Hydro.

From 1953 until 1991, another electrolysis-based ammonia plant was operated, in Glomfjord, producing about 330 t-NH$_3$ d$^{-1}$. The plant used the same electrolyzers as at

Norsk Hydro's Rjukan site [35,38]. Due to the emergence of low-cost natural gas in Norway, this plant was closed down in 1991.

### 3.2. Egypt

The first renewable ammonia plant in Egypt, and probably the country's first synthetic ammonia plant, was built in 1960. The plant was near the city of Aswan, close to a hydroelectric power station on the Aswan River. The hydroelectric power station and the electrolyzers of the renewable ammonia plant are shown in Figure 8. The reason for building the renewable ammonia plant in 1960 was national food security and the absence of natural gas, which became commercially available in Egypt only from 1970 onwards [39].

With a production of 400–500 t-NH$_3$ d$^{-1}$ or 140–175 kt-NH$_3$ y$^{-1}$, the ammonia plant in Aswan was the largest renewable ammonia plant ever built. Initially, the plant was operated with electrolyzers from De Nora, but these were replaced with electrolyzers from Brown Boveri in 1977 [40].

This renewable ammonia plant, also known the KIMA plant, was still operational into the 2000s [41]. However, a second ammonia plant was recently built in Aswan for the production of urea, based on natural gas. This fossil-based ammonia plant replaced the renewable ammonia plant, which was closed down in 2019 [42].

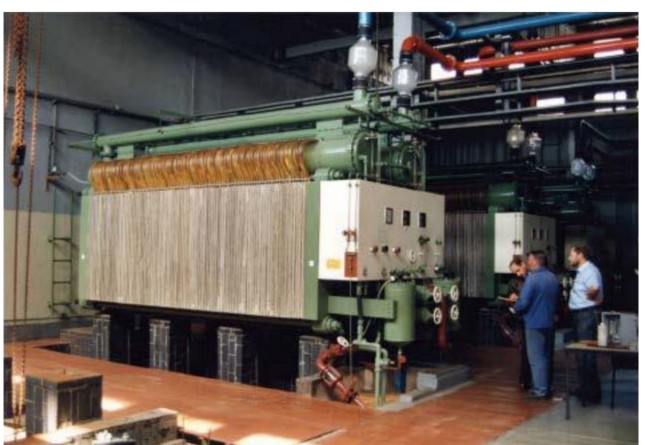 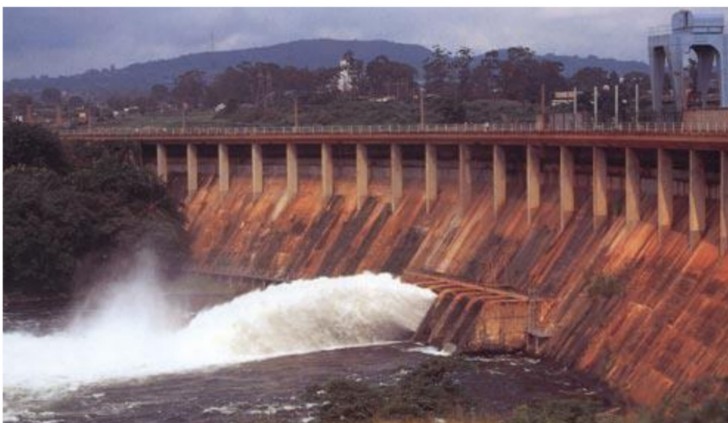

**Figure 8.** Left: Electrolyzers in the Aswan ammonia factory in Egypt, constructed in 1960. Reproduced from reference [43]. Right: Aswan Dam. Reproduced from reference [43]. Courtesy University of Michigan.

### 3.3. India

In 1958, a renewable ammonia plant became operational in Nangal, located in the northern part of India. The electricity was derived from the Bhakra Dam.

The plant produced about 400 kt-CAN y$^{-1}$ (calcium ammonium nitrate), equivalent to about 285 t-NH$_3$ d$^{-1}$ or 100 kt-NH$_3$ y$^{-1}$ [44]. Itwas equipped with De Nora electrolyzers.

The electrolyzers were operated at a reduced load from 1978 onwards, when the ammonia plant was changed to fuel oil as feedstock [45]. The plant was converted to natural gas in 2013 [46].

### 3.4. Peru

A renewable ammonia plant was built in Cusco, Peru, in 1962. By 1965, the plant had a rated capacity of 5200 Nm$^3$ H$_2$ h$^{-1}$ from seven parallel electrolyzers with a rated capacity of 3.5 MW, equivalent to about 60–65 t-NH$_3$ d$^{-1}$ or 20 kt-NH$_3$ y$^{-1}$ [31]. These electrolyzers are of the high-pressure bipolar-electrolyzer type, operating at 90 °C and 30 bar, manufactured by Lurgi.

The Cuzco plant is the only renewable ammonia plant that is currently still in operation. The plant is equipped with ThyssenKrupp electrolyzers, operating at an energy

consumption of $\leq 4.3$ kWh Nm$^{-3}$ H$_2$ [47]. A typical module of 10 MW consists of three stacks of 3.3 MW. The hydrogen pressure at the electrolyzer outlet is about 0.3 bar. Assuming an energy consumption of about 4 GJ t-NH$_3$$^{-1}$ for nitrogen purification and ammonia synthesis [48], this gives a total energy consumption of about 34 GJ t-NH$_3$$^{-1}$.

### 3.5. Zimbabwe

The last electrolysis-based ammonia plant to be built, during 1972–1974, is at Kwekwe, Zimbabwe (see Figure 9). By 1975, the plant produced about 72.6 kt-NH$_3$ y$^{-1}$. Hydroelectric power was delivered from the Kariba Dam. Twenty-eight alkaline electrolyzers with a rated capacity of 3.5 MW were operated in parallel, based on high-pressure bipolar electrolyzers manufactured by Lurgi. The total rated capacity of these electrolyzers was about 21,000 Nm$^3$ H$_2$ h$^{-1}$. The electrolyzer design was the same as for the Cusco plant in Peru, e.g., operating at 90 °C and 30 bar. In 2015, the plant was decommissioned due to an energy deficit in Zimbabwe and the associated high electricity prices.

Given that the plants in Peru and Zimbabwe used similar electrolysis technology, the energy consumption of these plants is probably similar. The energy consumption of the electrolyzers was about 4.5 kWh Nm$^{-3}$-H$_2$ [20], equivalent to about 32 GJ t-NH$_3$$^{-1}$. The nitrogen purification and ammonia synthesis add another 4 GJ t-NH$_3$$^{-1}$ [48], yielding a total energy consumption of about 36 GJ t-NH$_3$$^{-1}$.

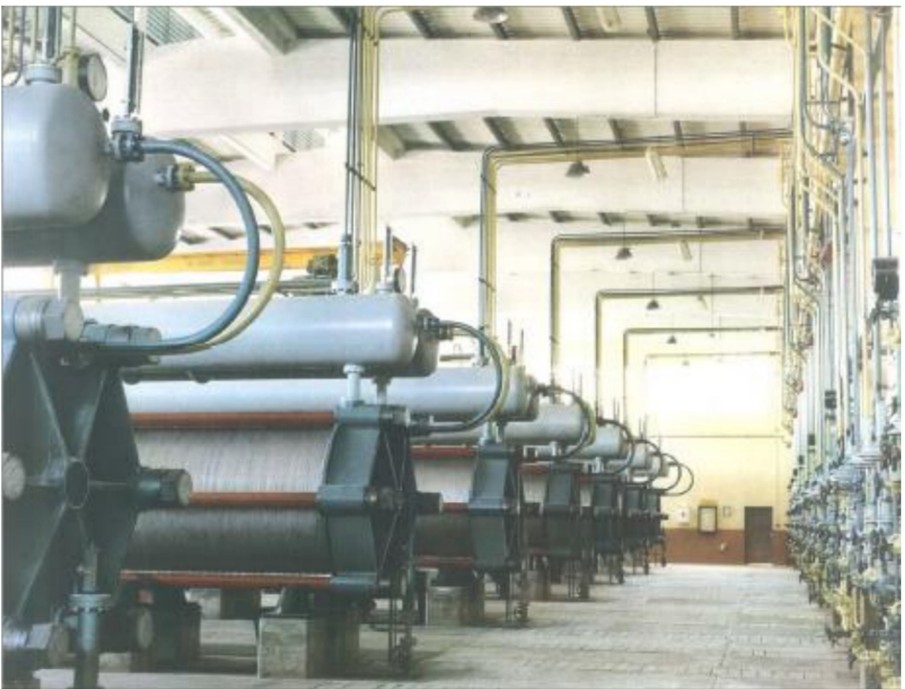

**Figure 9.** *Cont.*

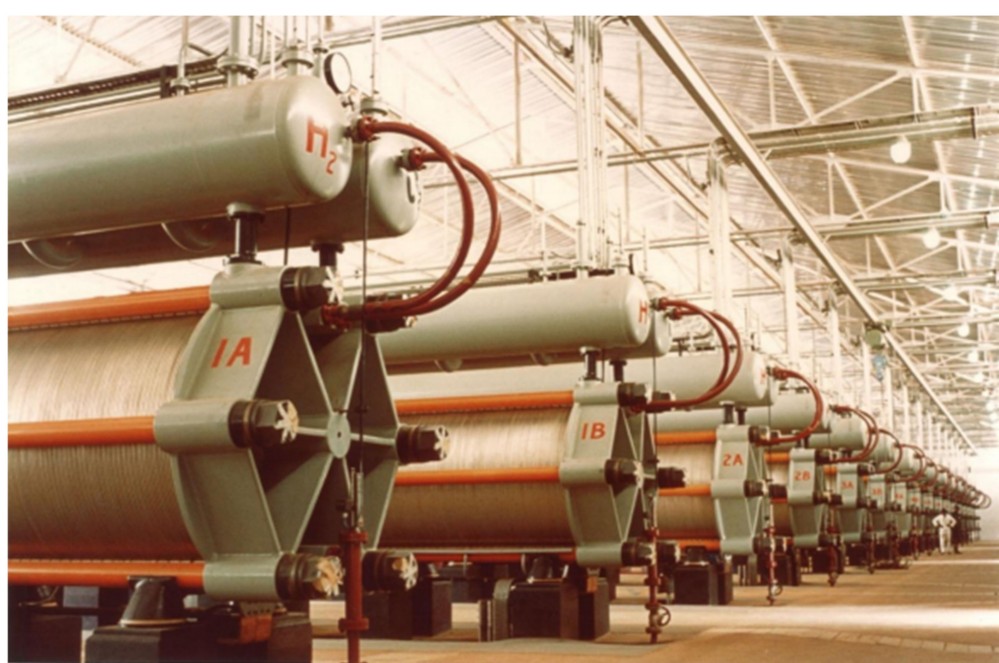

**Figure 9.** Top: Electrolyzers at the Cuzco ammonia production facility. Reproduced from reference [43]. Bottom: Electrolyzers at the Kwekwe ammonia production facility. Reproduced from reference [43].

## 4. 1960s–2021: Natural Gas Outcompetes Renewable Ammonia Production on a Large Scale

During the 1960s, renewable ammonia production reached its peak with an annual production of 0.65 Mt (see Figure 6). This represented about 4% of global ammonia production. Only one renewable plant was built after the 1960s, namely that at Kwekwe, in Zimbabwe (see Section 3.5).

The reasons for the decline of renewable ammonia synthesis in the latter half of the 20th century are discussed in this section. Especially, the emergence of abundant and low-cost natural gas was responsible for the decline in renewable ammonia production. Currently, essentially all ammonia production outside China is based on natural gas [7]. The four main reasons for the decline of renewable ammonia synthesis in favor of natural gas-based ammonia synthesis are:

- Technology improvements for fossil-based hydrogen production, especially for natural gas-based hydrogen production (Section 4.1);
- Cost reductions and availability of fossil-based feedstocks, especially natural gas (Section 4.2);
- Better cost-scaling of fossil fuel-based technologies (Section 4.3);
- Globalization of the fertilizer trade (Section 4.4).

### 4.1. Technology Improvements for Fossil Fuel-based Hydrogen Production

The technology for large-scale natural gas-based hydrogen production was introduced for ammonia synthesis during the 1940s–1950s. This began with the United States wartime program, when six new facilities adopted the primary and secondary steam-reforming of natural gas (methane), a technology developed by ICI [15]. This was followed with new markets for pure hydrogen, namely for hydrotreating in refineries. A benefit of natural gas over coal-based hydrogen production is the lower sulfur content in natural gas, requiring less clean up at the front end. Since then, the technology for natural gas-based hydrogen production has substantially improved. Initially, the energy consumption for natural gas-based ammonia synthesis was 55 GJ t-$NH_3^{-1}$, while the lowest current energy consumption is currently 26 GJ t-$NH_3^{-1}$ [5]. The reasons for this decrease in energy

consumption include, for individual high-capacity ammonia units of over 600 t-NH$_3$ d$^{-1}$, the introduction of pressure reforming and centrifugal compressors (replacing reciprocating compressors), which enabled a tripling of the capacities of the individual converters. Improved heat integration through process optimization enabled the scale-up in single-train energy-integrated ammonia units. In addition, there were advances in improved catalyst activity for the lower temperature and pressure operations which accompanied the scale-up [5]. There is an extensive literature describing historical developments in fossil-based ammonia production [49–52].

In contrast, the technology for electrolysis-based hydrogen production has remained remarkably unchanged. Currently, the most efficient electrolysis-based ammonia synthesis technology consumes about 33 GJ t-NH$_3$$^{-1}$ [7], compared with about 50 GJ t-NH$_3$$^{-1}$ a century ago [11]. An indication of the changes in energy consumption for coal-based ammonia synthesis, electrolysis-based ammonia synthesis, and natural gas-based ammonia synthesis is shown in Figure 10.

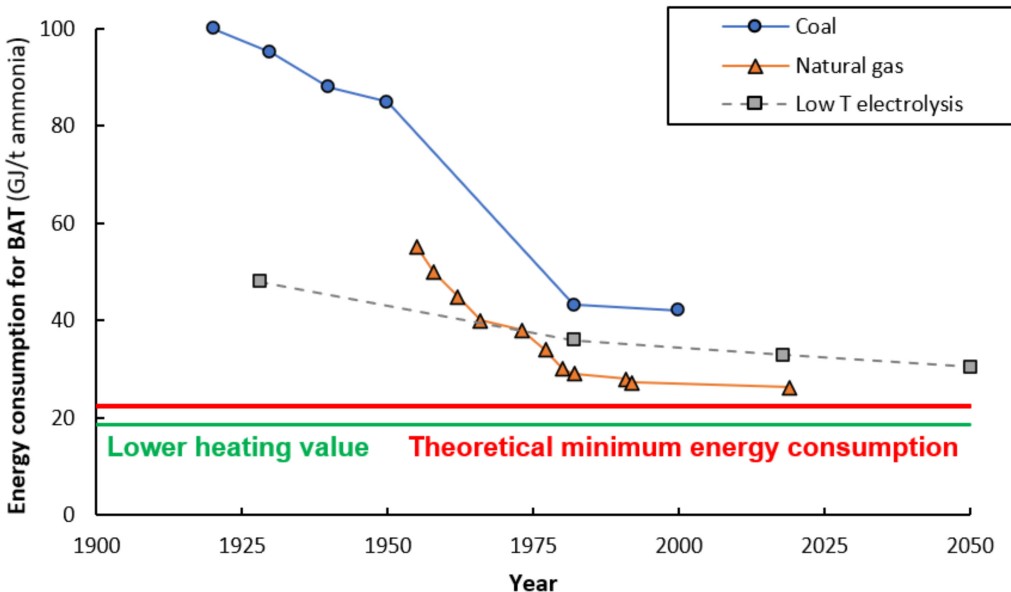

**Figure 10.** Technology-based development for coal-based ammonia synthesis, electrolysis-based ammonia synthesis, and natural gas-based ammonia synthesis. Partially adapted from references [5,7]. BAT: best available technology.

### 4.2. Cost Reductions of Fossil Feedstocks

Though in the late 1920s electricity was more costly than coal as a source of power, in the case of ammonia synthesis the energy consumption from coal was substantially higher than for electrolysis-based ammonia production (see Figure 10). This made renewable ammonia production competitive except for in the large plants (IG Farben, ICI, Du Pont, and Allied Chemical). However, because the energy consumption of coal-based ammonia synthesis, and especially natural gas-based ammonia synthesis, decreased after the mid-1940s (see Figure 10), electrolysis-based ammonia production became less competitive. This was notably the case from the mid-1960s following the introduction of the novel single-train high-capacity energy-saving units by the engineering contractor M.W. Kellogg.

Hydropower costs at least 30 USD/MWh [53,54], equivalent to about 8.3 USD/GJ or 7.9 USD/MMBtu, resulting in an electricity price of at least 300 USD/t for ammonia production. This does not include other expenditures, such as capital investment and operational costs, which also easily add 200 USD/t [55]. This is not competitive, given that ammonia market prices have typically been 200–300 USD/t. For comparison, the cost of natural gas can be as low as 2 USD/MMBtu in various locations, equivalent to an ammonia feedstock cost of 54 USD/t ammonia [51].

*4.3. Better Cost-Scaling of Fossil Fuel-Based Technologies*

During the 1920s, most ammonia synthesis plants were small, typically with a capacity < 100 t-NH$_3$ d$^{-1}$, with few exceptions [22]. At this process scale, the investment cost for a coal-based ammonia plant and an electrolysis-based ammonia plant was similar at about 4560 USD/t-NH$_3$/y and 4940 USD/t-NH$_3$/y, respectively in 2020 USD equivalent [10].

During the early 1960s, the largest individual fossil-based ammonia units had reported capacities of about 455 t-NH$_3$ d$^{-1}$ [15]. However, typical unit sizes at the time were 250–300 t-NH$_3$ d$^{-1}$. Several loops, or trains, each one a single ammonia synthesis unit, typically operated side by side to increase the capacity of a factory.

The largest renewable ammonia factory to date was also built during the 1960s, in Aswan, Egypt, with a capacity of 400–500 t-NH$_3$ d$^{-1}$. Since the 1960s, the size of typical fossil-based ammonia units have been scaled-up substantially, as mentioned above, due to technological developments and economies of scale [5]. With scale-up, more heat integration was possible, resulting in a lower overall energy consumption (Figure 10).

Currently, typical ammonia units have a capacity of 1000–3300 t-NH$_3$ d$^{-1}$ [15]. The largest fossil fuel-based ammonia synthesis unit currently in operation has a capacity of 3670 t-NH$_3$ d$^{-1}$ [56]. A historical overview of ammonia unit capacities is shown in Figure 11.

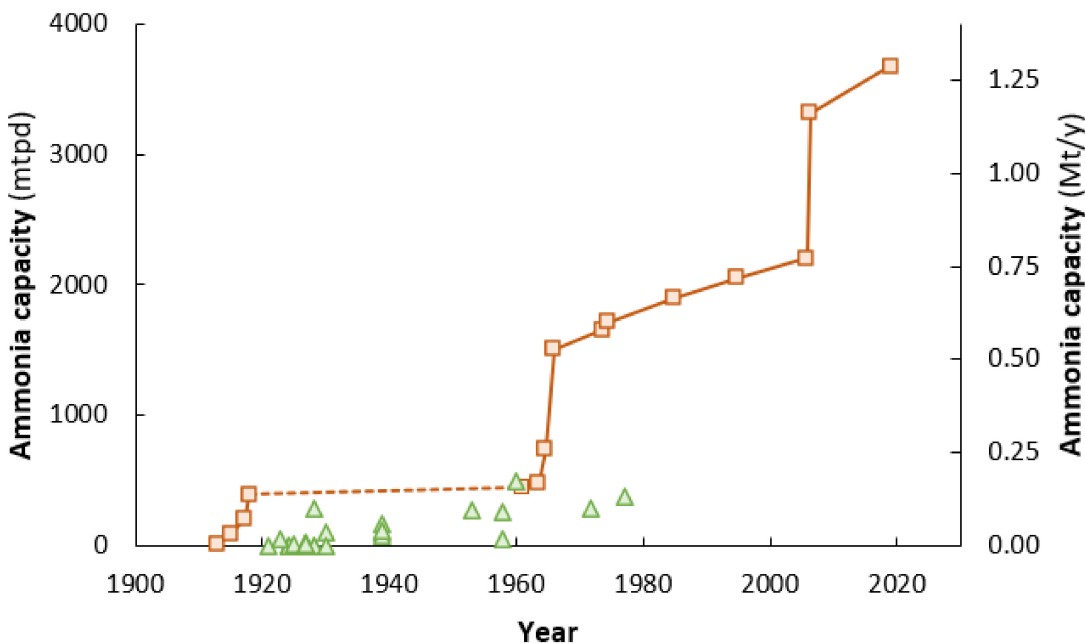

**Figure 11.** Ammonia production, individual unit capacities, through the years, fossil-based ammonia production (brown squares), and renewable ammonia production (green triangles). Estimates for the largest fossil-based ammonia units between 1913 and 1918 are from Travis [12], and between the mid-1960s and 2019 from Brightling [15] and ThyssenKrupp [56]. Estimates for renewable ammonia units are discussed in this article.

The cost of fossil-based technologies for hydrogen production scales substantially, with cost-scaling factors typically of 0.6. In contrast, electrolyzers are produced in modules with a few MW, which are increased in cases of higher hydrogen demand. Thus, the cost benefit of large-scale production is less substantial than for fossil-based hydrogen production technologies [35,57]. The cost-scaling for the ammonia synthesis loop still applies for renewable ammonia production [58]. The typical investment requirements of various ammonia synthesis technologies are shown in Figure 12.

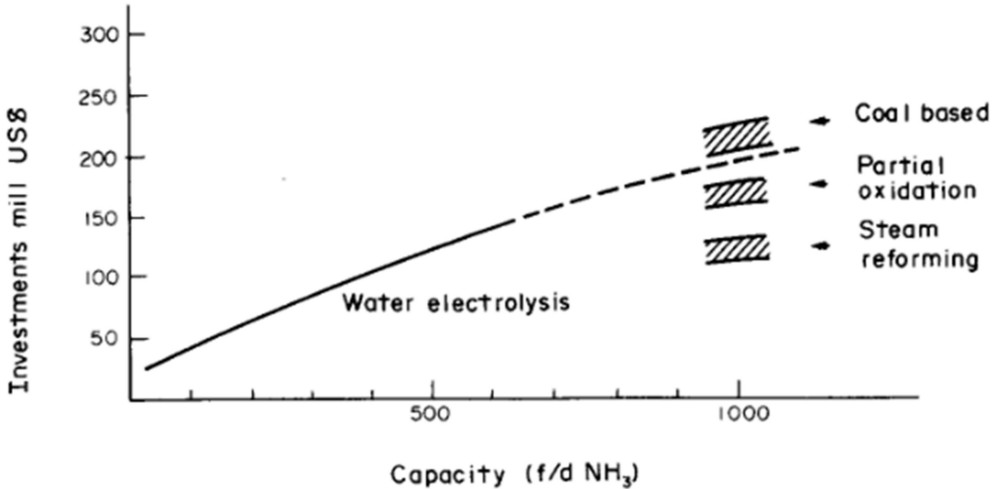

**Figure 12.** Indicative investment cost of various ammonia synthesis technologies. Reprinted with permission from Ref. [35]. 1982, Elsevier.

### 4.4. Globalization of the Fertilizer Trade

During the 1920s, renewable ammonia was manufactured to provide national food security, especially in Italy and Japan. Ammonia and derived fertilizers increasingly became global commodities in the decades that followed. Given that fossil-based ammonia production is often more cost-competitive (especially at large scales) than renewable ammonia production, and that the cost of ammonia transport is typically low, renewable ammonia was eventually unable to compete in the global ammonia fertilizer market [7,59]. In addition, urea, a fertilizer with the chemical formula $CO(NH_2)_2$, accounts for 55% of current ammonia utilization [6]. Urea synthesis requires a carbon feedstock, such as natural gas or coal, which is processed to yield synthesis gas (hydrogen and carbon monoxide), which is the source of carbon dioxide (from carbon monoxide by the shift reaction), that is reacted with ammonia. Decarbonized carbon feedstocks are currently expensive.

Recent events such as the COVID-19 pandemic and extreme climatic conditions reduced ammonia production in 2020, resulting in major supply problems for natural gas and high fertilizer prices in late 2021, and even more in early 2022 [60]. In light of this supply problem, a project has recently been announced in Kenya for the domestic production of a renewable fertilizer from renewable solar and wind energy [61]. This food security situation is similar to that of Italy and Japan during the 1920s (see Sections 2.2 and 2.4). Providing domestic food security and thus economic security may 'outcompete' or at least rival the global market again in certain cases.

## 5. 2021 and Beyond: Renewed Interest in Renewable Ammonia

During the 1980s, in the wake of the oil crises, ammonia gained interest as an energy vector in the hydrogen economy [62,63]. The interest in ammonia as an energy vector re-emerged in the early 2000s, due to growing concerns regarding the environmental impact of fossil fuels. Various authors have discussed the central role of ammonia in the hydrogen economy [64,65]. In 2004, the NH$_3$ Fuel Association (now the Ammonia Energy Association) organized its first NH$_3$ Fuel Conference in West Des Moines, Iowa. Since then, conferences on ammonia as a decarbonized energy vector and as a hydrogen carrier have been organized at least annually, with activities intensifying in recent years.

Especially since 2020, the momentum towards renewable ammonia has been substantial, with various world-scale renewable ammonia plants being announced for the present decade [7]. Several authors have reviewed the central role of ammonia in a hydrogen economy [59,66–70]. Furthermore, a consortium of industrial companies has indicated that ammonia will play a central role in decarbonizing the shipping sector [71]. Key factors for the renaissance of renewable ammonia production include: (1) increasing carbon

emission penalties, (2) the decreasing cost of renewable electricity from solar and wind, (3) the decreasing cost of electrolyzers and scale-up of electrolyzer capacity, and (4) the development of novel electrolysis technologies [7,21,72]. In fact, Saygin and Gielen [73] estimated a renewable ammonia production of 495 Mt by 2050 in their 1.5 °C scenario, which is nearly three times the current global ammonia production.

Novel approaches to electricity-driven nitrogen fixation are currently extensively researched, including electrochemical ammonia synthesis [67,74–77] and plasma-catalytic ammonia synthesis [78–81]. Such technologies can follow the load fluctuations of renewable electricity, and are more easily scaled down than an electrolysis-based Haber–Bosch plant. However, scientific challenges remain for these technologies [82–84], and the economics are not (yet) competitive with an electrolysis-based Haber–Bosch process [85–87].

A century ago, in 1921, renewable ammonia production stimulated not only the global production of synthetic ammonia but also the development and scale-up of the electrolyzer industry. Today, the electrolyzer industry is once more spurred on by the development of large-scale renewable ammonia projects [7].

A current technological challenge is the fluctuation in renewable electricity from solar and wind. The typical solution for this is oversizing the electrolyzers for hydrogen production, and storing compressed hydrogen [88]. This allows the operation of the ammonia synthesis loop with minimal fluctuations. Such a facility is already under construction in Puertollano, Spain, where fossil-based hydrogen production was partially replaced by renewable hydrogen from solar PV electricity, a battery, water electrolysis, and compressed hydrogen storage [89].

Alternatively, the natural gas feedstock of a gas-based ammonia plant can be replaced with, for instance, biogas, to decrease the carbon footprint of the ammonia product [7]. Furthermore, biomass can be gasified for hydrogen production, with the subsequent conversion to ammonia [90].

In conclusion, renewable ammonia is set to make a major comeback and play a key role in the decarbonized energy landscape and in the hydrogen economy. While renewable ammonia played a key role in national food security for countries without fossil resources during the 20th century, it promises, in addition, national and global energy security during the 21st century.

**Supplementary Materials:** The following supporting information can be downloaded at: https: //www.mdpi.com/article/10.3390/suschem3020011/s1, Table S1: estimated ammonia production capacity by feedstock [3,6,54,91–97]; Table S2: reported renewable ammonia production capacities, apart from Italy [22,31,34,35,38,40,41,98–105]; Table S3: industrial alkaline electrolyzers [31,34,106].

**Author Contributions:** Conceptualization, K.H.R.R.; formal analysis, K.H.R.R.; investigation, K.H.R.R.; resources, K.H.R.R. and A.S.T.; writing—original draft preparation, K.H.R.R.; writing—review and editing, K.H.R.R., A.S.T. and L.L.; visualization, K.H.R.R.; supervision, A.S.T. and L.L.; project administration, K.H.R.R.; funding acquisition, L.L. All authors have read and agreed to the published version of the manuscript.

**Funding:** This research was co-financed by TKI—Energie from Toeslag voor Topconsortia voor Kennis en Innovatie (TKI) from the Ministry of Economic Affairs and Climate Policy, The Netherlands.

**Institutional Review Board Statement:** Not applicable.

**Informed Consent Statement:** Not applicable.

**Data Availability Statement:** Not applicable.

**Conflicts of Interest:** The authors declare no conflict of interest.

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
