# Peer review of "1921–2021: A Century of Renewable Ammonia Synthesis"

_2673-4079, doi:10.3390/suschem3020011_

Round 1

Reviewer 1 Report

The topic of the presented manuscript is interesting and up to date. It fits the current research trend towards using ammonia as a hydrogen carrier. Nowadays, the development of the hydrogen economy makes renewable ammonia synthesis more and more important. The manuscript would be of interest to many scientists working in ammonia production, especially those focused on the development and modifications of this process, which is currently an essential issue in adapting ammonia as an energy vector. Moreover, as the Authors emphasized, there is no comprehensive review discussing the history of the renewable ammonia synthesis process in the known literature.

The manuscript is well and clearly written. The presented information is well organized, and the described history is well presented and supplemented with photos and figures with statistical data. I did not have access to the Supporting Information. Nevertheless, it did not make it challenging to analyze the text or understand the topic presented. In my opinion, the paper can be published in Sustainable Chemistry after a minor revision:

  • Minor editing of the text is necessary - double spaces, duplicate words, etc., can be found in the text. Please also check the Figure captions and correct them - some did not provide references from where the photos were obtained (e.g. Figs. 8 and 9).
  • Please check the Reference list carefully – some bibliographic data seem incomplete and may make it difficult to find these references.

Author Response

We thank reviewer 1 for the fair feedback and punctuality. We have made changes to the manuscript in accordance with the reviewer comments.

We have removed the double spaces from the manuscript.

The references of Figs. 8 and 9 were added.

Furthermore, we have updated the reference list, such that all references are complete (e.g., links were added to the reports).

Reviewer 2 Report

This review by Rouwenhorst et al. is an interesting and timely paper on an important topic, ammonia synthesis from renewable resources. It covers the history of renewable ammonia synthesis from 1921 to 2021. The article is worthy of publication, but a few minor details need to be addressed. 

1. The text in Figure 2 is not legible. I understand that the figure was reprinted from a publication in the 1920s. However, the authors are encouraged to relabel the equipment clearly for the benefit of the reader.

2. In Table 2, the authors used "????" for the year of the San Giuseppe di Cairo site. They are encouraged to use "N.A." or some notations that are more professional.

3. The authors mentioned the economics of renewable ammonia synthesis techniques such as electrolysis. At the conclusion of the article, it would be good to also highlight significant recent technological advances for renewable ammonia synthesis using electrochemical and plasma-related methods and how these technological advances may enable the further development and employment of renewable ammonia synthesis. 

4. Also, I do not have access to the Supplementary Materials. Please make sure that it is available.

Author Response

We thank reviewer 2 for the valuable feedback. We have made changes to the manuscript in accordance with the reviewer comments.

  1. The text in Figure 2 has been edited to make the figure better readable. See below:

2 .The ‘????’ in Table 2 has been changed to ‘N/A’.

  1. It is indeed good to mention advances in science. Therefore, a paragraph was added to discuss the status of electricity-driven nitrogen fixation. See section 5 (page 18):

“Novel approaches to electricity-driven nitrogen fixation are currently extensively re-searched, such as electrochemical ammonia synthesis [67, 74–77] and plasma-catalytic ammonia synthesis [78–81]. Such technologies can follow load fluctuations of renewable electricity, and are more easily scaled down than an electrolysis-based Haber-Bosch plant. However, scientific challenges remain for these technologies [82–84], and the economics are not (yet) competitive with an electrolysis-based Haber-Bosch process [85–87].”

  1. The Supplementary Materials is now made available.

Reviewer 3 Report

Journal : Sustainable Chemistry (ISSN 2673-4079)

Manuscript ID: suschem-1621336

Type: Review

Title: 1921-2021: A Century of Renewable Ammonia Synthesis

Authors: Kevin H. R. Rouwenhorst * , Anthony S. Travis , Leon Lefferts

Reviewing report

The submitted manuscript consists of a description of operating processes over the last hundred years in the world for the synthesis of ammonia using a renewable source of hydrogen in the feedstock. The authors mainly associated the renewable specification of the ammonia process with the production of hydrogen using mainly electrolysis and hydropower.

  • My first question: what criteria do the authors take into account to categorize the ammonia synthesis process as renewable? is it applicable to all stages of the process, including the use of ammonia?

If it's just related to hydrogen I suggest switching to "sustainable" or "ammonia synthesis from renewable hydrogen"

  • The review is described in several sections, including the geographical distribution of world ammonia production. The authors did not address the very important aspect related to thermodynamic analysis and kinetics as key steps in the evaluation of the different processes highlighted. Provide the necessary data at these levels.

  • Technological barriers should be discussed further. The question is how a process using hydrogen from various renewable sources (not just hydrogen from electrolysis) can be integrated into a conventional ammonia synthesis process?

  • Schematic diagram for flow circulation in fig. 2 need improvement.

The review in my opinion is interesting, it needs to be restructured and enriched before being considered for publication in Sustain. Chem. Journal.

Author Response

We thank reviewer 3 for the valuable feedback, which helps to improve the content of this review. We have made changes to the manuscript in accordance with the reviewer comments.

With regards to the nomenclature ‘sustainable’ vs. ‘renewable’, it is important to mention that hydrogen production accounts for over 90% of the energy consumption of an ammonia synthesis plant. Furthermore, equipment such as compressors can be operated with electricity.

This is now mentioned in section 2.1 (page 5):

“It should be noted that hydrogen production typically accounts for more than 90% of the required energy input. Furthermore, the compressors for feed gas compression and re-cycling of unreacted gas in the ammonia synthesis loop can be operated with renewable electricity. If renewable electricity is the economical method for hydrogen production, it will also be favourable for gas compression and recirculation in the synthesis loop. Thus, ammonia production using renewable hydrogen results in renewable ammonia. It should be noted that the implicit assumption that the produced ammonia converts completely back to N2 is reasonable when using ammonia as energy carrier.”

A high-level thermodynamic analysis and kinetic reasons for the operation conditions was added to the manuscript, section 2.1 (page 5):

“From a thermodynamic point of view, ammonia synthesis benefits from a low temperature and a high pressure. However, the H2 and N2 do not spontaneously react to form ammonia, unless the temperature is increased to several thousand Kelvin. This is impracticable in industry. Therefore a catalyst is essential to increase the ammonia synthesis rate for industrial application.

All synthetic ammonia technologies developed during the 1920s relied on a multi-component iron-based catalyst, high temperatures and high pressures (400-650°C and 200-1000 bar), and ammonia removal by condensation. The exact formulation of the iron-based catalyst varied among ammonia synthesis processes, because most companies developed their own catalysts due to a lack of international collaboration [12]. In this connection there was an element amount of industrial espionage.

The catalyst formulation has a major impact on the activity. A less active catalyst re-quires a higher operating temperature to achieve sufficient activity for ammonia formation. However, a higher temperature is not beneficial for the equilibrium as explained above. Thus, the pressure is increased to improve the equilibrium conversion to ammonia.

Most processes, including the original Haber-Bosch process, required refrigeration to sub-atmospheric temperatures to produce liquid, anhydrous ammonia. However, the Casale and Claude processes did not require such refrigeration to sub-atmospheric temperatures to produce liquid, anhydrous ammonia due to their very high operating pressures (500-1000 bar). Thus, the operating pressure influences both the thermodynamic equilibrium and the liquefaction temperature. It should be noted that increasing the pressure increases the energy requirement for compression of the hydrogen and nitrogen feed-stock.”

Technological barriers have been added for new ammonia plants, such as fluctuating renewable electricity. Furthermore, alternative gas feedstocks have been added, see section 5 (page 19):

“A current technological challenge is the fluctuation in renewable electricity from solar and wind. The typical solution for this is oversizing the electrolyzers for hydrogen pro-duction, and storing compressed hydrogen [88]. This allows the operation of the ammonia synthesis loop with minimal fluctuations. Such a facility  is already under construction in Puertollano, where fossil-based hydrogen production was partially replaced by renew-able hydrogen from solar PV electricity, a battery, water electrolysis, and compressed hydrogen storage [89].

Alternatively, the natural gas feedstock of a gas-based ammonia plant can be re-placed with for instance biogas to decrease the carbon footprint of the ammonia product [7]. Furthermore, biomass can be gasified for hydrogen production, with subsequent conversion to ammonia [90].”

Figure 2 has been updated to make the figure better readable.

Round 2

Reviewer 3 Report

The new version prepared by authors is mostly  in accordance with what was required except for thermodynamic analysis. The comment appearing in page 5  are basic and not sufficiently  developed.